

# 1 Classification and susceptibility assessment of debris flow

# 2 based on a semi-quantitative method combining of the fuzzy

# 3 C-means algorithm, factor analysis and efficacy coefficient

Zhu Liang[1], Changming Wang[1], Songling Han[1], Kaleem Ullah Jan Khan[1] and Yiao Liu[1]
(College of Construction Engineering, Jilin University, 130000 Changchun, People's Republic of
China)
Correspondence to: wangcm@jlu.edu.cn
Abstract: The existence of debris flows not only destroys the facilities, but also seriously threatens
human lives, especially in scenic areas. Therefore, the classification and susceptibility analysis of
debris flow are particularly important. In this paper, 21 debris flow catchments located in Huangsongyu
town ship, Pinggu District of Beijing, China were investigated. Besides field investigation, geographic
information system, global positioning system and remote sensing technology were applied to
determine the characteristics of debris flows. This article introduced clustering validity index to
determine the clustering number, and the fuzzy C-means algorithm and factor analysis method were
combined to classify 21 debris flow catchments in the study area. The results were divided into four
types: scale-topography-human activity closely related, topography-human activity-matter source
closely related, scale-matter source-geology closely related and topography-scale-matter source-human
activity closely related debris flow. And 9 major factors screened from the classification result were
selected for susceptibility analysis, using both the efficacy coefficient method and the combination
weighting. Susceptibility results showed that the susceptibility of 2 debris flows catchments were high,
6 were moderate, and 13 were low. The assessment results were consistent with the field investigation.
Finally, a comprehensive assessment including classification and susceptibility evaluation of debris
flow was obtained, which was useful for risk mitigation and land use planning in the study area, and
provided reference for the research on related issues in other areas.
**Keywords** Debris flow classification, Susceptibility, Fuzzy C-means algorithm, Factor analysis,
Efficacy coefficient method

# 29 1 Introduction

Debris flow is a common geological disaster widely distributed across the world. Due to its sudden
outbreak, it is often difficult to give real-time warning. Debris flow usually flows at a speed of 0.8-28
tn/s (Dieter et al., 1999; Clague et al., 1985), inflicting severe damage to lives and properties once it
occurs. China is one of the worst affected areas prone to natural disasters. According to data, there are
nearly 8,500 debris flows distributed across 29 provinces, with an area of approximately $4.3 \times 106$ km$^2$
(Ni et al., 2010). Every year, nearly one hundred counties are directly endangered by debris flow, and





36 hundreds of people lose their lives, resulting in irreparable losses (Kang et al., 2004).

37   Debris flow susceptibility analysis (DFS), which expresses the likelihood of a debris flow

38 occurring in an area with respect to its geomorphologic characteristics (Blais et al., 2016), is very

39 important to mitigate, evaluate and control debris flow disasters (Chiou et al., 2015). Physical,

40 empirical, and statistical approaches are used to analyze debris flow, which expresses the presumption

41 of a debris flow occurring in an area with respect to its geomorphologic characteristics (Blais et al.,

42 2016). Physical-based approaches (Carrara et al., 2008; Burton and Bathurst, 1998) are more applicable

43 to analyze physical and mechanical factors in independent catchments. Empirical model belongs to

44 qualitative evaluation and is too subjective to be convinced. Statistical analyses which are usually

45 applied in the research of regional debris flow, belongs to quantitative evaluation and depends on the

46 completeness and accuracy of data. For a study area with a limited number of debris flows, a

47 semi-quantitative evaluation method is more appropriate. This analysis includes the extraction of

48 evaluation factors, the determination of weight factors and the establishment of an evaluation model.

49 Considering that the influencing factors of debris flow are complex, multiple evaluation indexes are

50 generally involved, and linear correlations between different factors further complicate debris flow

51 susceptibility analysis (Benda et al., 1990). However, the unreasonable selection of factors may cause

52 the loss of important information and failure to obtain accurate evaluation results. One way to alleviate

53 these problems is dimension reduction through exploratory factor analysis (Aguilar et al., 2000). Some

54 researchers (Peggy et al., 1991; Ming et al., 2016) have used the principal component analysis method

55 to conduct effective dimensionality reduction for selected factors and eliminate the correlation between

56 factors. However, the coefficient of principal component after dimensionality reduction can be positive

57 or negative, which is not ideal for which is not ideal for the occurrence of debris flow. Factor analysis,

58 in which the coefficients of the common factors are all positive, and the variables are more resolvable

59 by rotation technology is applied in the current study.

60   To determine the influence of different factors on debris flow susceptibility, the weights of these

61 factors should be assigned first. The combined weighting method, which possesses the advantages of

62 subjective and objective weighting methods, was applied to assign factors with logical weights.

63   The efficiency coefficient method (ECM) is a comprehensive evaluation method based on

64 multiple factors and is suitable for complex research objects, such as debris flow. The factors can be

65 converted into measurable scores through the appropriate function and objectively reflect the situation

66 of the evaluation object in the case of a large difference in the factor value. This research primarily

67 focuses on the method, which is applied to the debris flow susceptibility evaluation based on the results

68 of the weight analysis.

69   Debris flow classification plays a direct guiding role in disaster prevention and mitigation, and

70 mature classification methods have been developed (Iverson et al., 1997; Brayshaw et al., 2009).

71 However, a single classification standard cannot fully and accurately reflect the comprehensive

72 characteristics of debris flow ditches, and base on different classification criteria, the same debris flow

73 will belong to different types at the same time. The fuzzy C-means (FCM) method which is applicable

74 to a wide variety of geostatistical data analysis (Bezdk et al., 1981), was applied to classify debris flow

75 in this paper. Considering that the main influencing factors of different types of debris flow are also

76 different, FA was carried out for each category to obtain major factors to define each type of debris

77 flow.

78   In recent years, with the improvement of computer performance and the advance features in

79 geographic information systems (GIS), global positioning systems (GPS) and remote sensing (RS)





techniques, also known as "3S technology", has become very effective and useful especially to debris
flow research (H. Ġomez 2008; Glade T 2005; Conway SJ 2010). In particular, the application of GIS
has greatly improved the ability of spatial data processing and analysis, such as slope direction analysis
and flow direction calculation (Mhaske et al., 2010; Xu et al., 2013; Kritikos et al., 2015). Therefore,
FA、FCM and ECM were used to classified and evaluated the susceptibility of debris flow in the current
study, combining with "3S technology" and field investigation.

## 2 Study area

The research area is located around several scenic spots in Huangsongyu township, Pinggu district,
Beijing. The village covers an area of 12.83 square kilometers, including 732 households, a total of
2043 people. And the Shilin gorge is the core scenic area of Huangsongyu geopark, attracting a large
number of tourists all year round. The geographical location of the study area and 21 debris flow
catchments are shown in Fig. 2. During our field investigation, some scenic spots have been closed
down due to the threat of falling rocks, floods and debris flow, which were shown in Fig.3. And Fig.4
and Fig.5 show the situation of the other two scenic spots, respectively. Considering the sudden and
rapid outbreak of debris flow and the large number of tourists and surrounding villagers in the scenic
area, it is necessary to assess the susceptibility of debris flow.
The study area is located in the northwest of north China plain, which belongs to yanshan
mountain range. Surrounded by high terrain, the central is flat, and the highest elevation of the territory
is 1188m, the lowest is 174m. The Yanshanian and Indosinian periods in the study area were
characterized by strong tectonic activity, which resulted in a series of large fold and fault structures.
Due to long-term geological processes, the structure in the area is relatively complex. But the strata are
relatively simple, except for a few Archean metamorphic rocks, the exposed strata are middle
Proterozoic sedimentary strata and Quaternary sediments. The main lithology of the Archean age (Ar)
is amphibious plagiarize gneiss and black cloud matinee. The Great Wall system (Ch) is the broadest
strata in this area, and the main lithology is dark gray ferric dolomite, sacrilegious micritic dolomite,
dolomite sandstone. The main lithology of jixian system (Jx) is dolomite. Quaternary system (Q) is
dominated by sand, gravel and clay of residual and diluvial facies. The non-developed lithology of
magmatite is mainly granite and quartz diorite.
The study area is characterized by a north temperate continental climate with distinct four seasons
and large annual temperature difference. The coldest average January temperature is 6 ~ 8 °C and the
hottest July average temperature is 21.6 °C. The annual precipitation is about 639.5mm, and the average
monthly rainfall (1959-2017) is shown in Fig. 1. And the precipitation in summer is the most,
accounting for 74.9% of the annual precipitation, which is generally concentrated in late July and early
August, promoting debris flow.

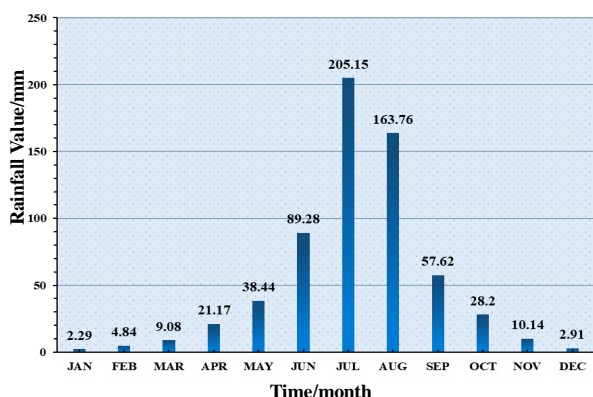


**Fig1.** Average monthly rainfall data (from 1959to2017) for Pinggu district

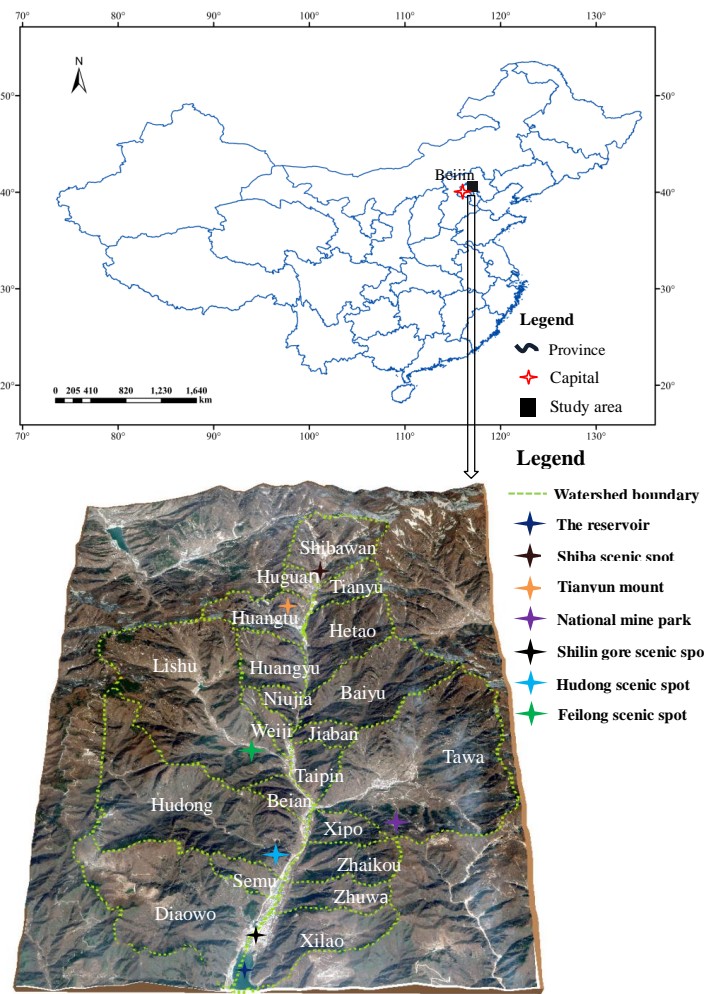



**Fig.2** Geographical positions of the Huangsongyu scenic region and the investigated 26 debris flow
catchments

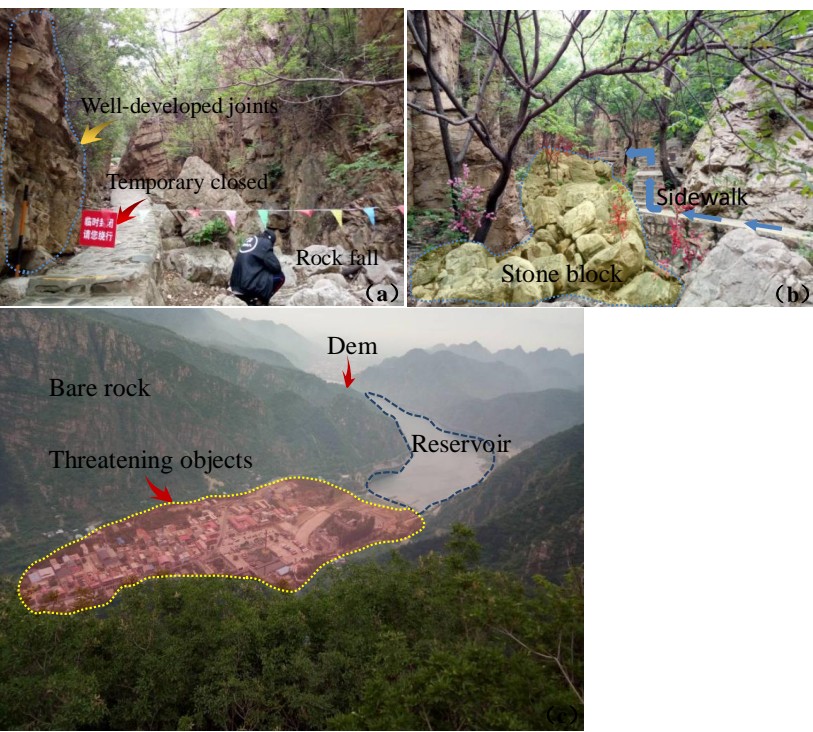


**Fig.3** Shilin gore scenic spot. **a** part of the scenic area have been closed, **b** the scenic area was heavily
blocked by rockfill, **c** threatening object of debris flow

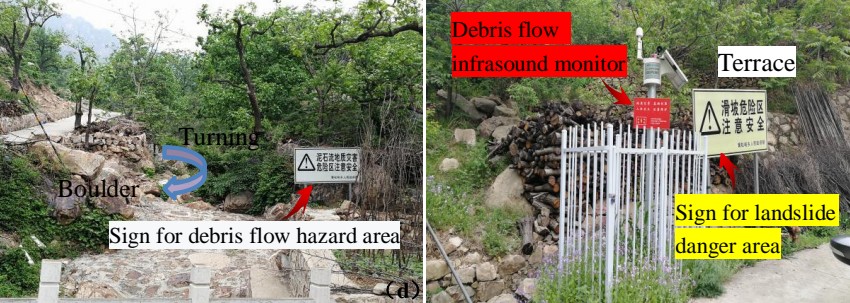







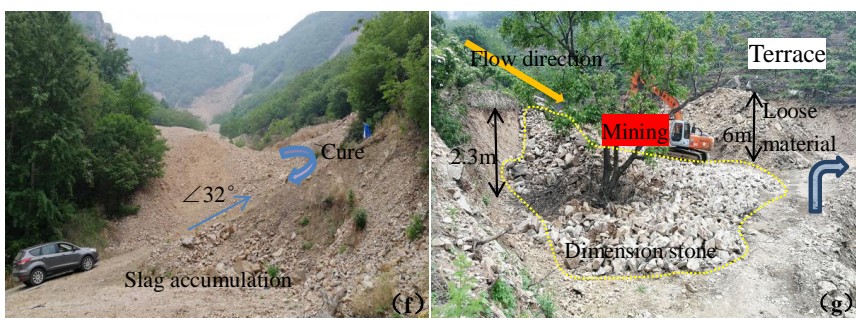

**Fig.4** Huangsongyu national mining park. d sign for debris flow hazard area, e debris flow monitoring
instrument, f loose slag accumulated in formation area, g excavator mining

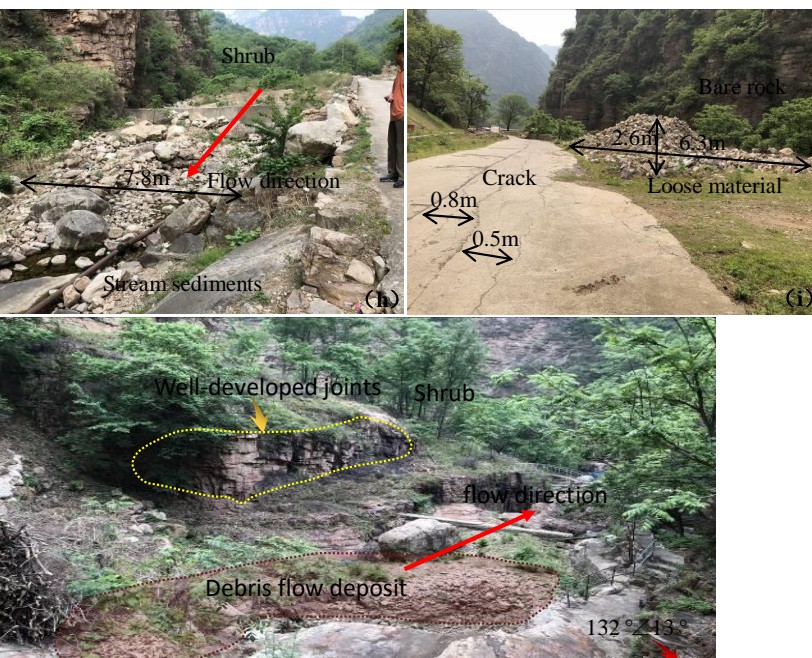

**Fig.5** Lishu scenic spot. h stream sediments, i road cracks, g debirs flow deposit.
## 3 Methodology
### *3.1 Fuzzy c-means clustering (FCM)*
The fuzzy c-means method belongs to soft clustering, which is widely used at present. Its core idea is




to map data points of multi-dimensional space to different clustering sets in the form of membership
degree, so as seeks C cluster centers in such a manner that the intercluster associations are minimized
and the intracluster associations are maximized (Bezdek et al., 1981). For every group, each point is
assigned a membership degree between 0 and 1. The membership values indicate the probability of
each point belonging to the different groups (Samuel et al., 2019). The steps of FCM algorithm are as
follows (Fig.6):
(1) The membership matrix u is initialized with random Numbers between 0 and 1, which is used to
represent the degree to which the object belongs to a set. And it satisfies the constraint conditions:
$$\sum_{i=1}^{C} \mu_{ij} = 1, j = 1,2......, n \tag{1}$$

(2) Calculating clustering centers $C_i$ and the formula is as follows (Hammah et al., 1998)
$$C_i = \sum_{j=1}^{n} u_{ij}^m x_j \Big/ \sum_{j-1}^{n} u_{ij}^m \tag{2}$$

where m controls the degree of fuzziness and m = 2 is deemed to be the best for most applications
(Bezdek et al., 1981).
(3) Determining the number of clustering centers
The clustering number C of FCM algorithm is not clearly given, which is one of the key factors
affecting the clustering effect. So this paper combines the non-distance-based FCM clustering
effectiveness index proposed by Chen and Pi (Chen et al., 2013) to determine the value of C. The
exponent(Vcs) consists of the degree of compactness and the degree of dispersion. And the definition
of compactness is as follows:
$$C_{ij} = \begin{cases} u_{ij}^2, u_{ij} \geq \dfrac{1}{c} \\ 0, u_{ij} < \dfrac{1}{c} \end{cases} \tag{3}$$

where Cij is the compactness of class i and class j samples. When uij is greater than or equal to 1/c, it
indicates that the membership degree of the Jth class samples belonging to the ith class is high. When
uij<1/c, it indicates that the J sample is unlikely to belong to the i th class. When all samples clearly
belong to a certain class, the compactness degree is the maximum. That is, the clustering result is
compact. Sum over the compactness between all samples and all classes and the formula is as follow:
$$C = \sum_{i=1}^{c} \sum_{j=1}^{n} C_{ij} \tag{4}$$

The definition of dispersion is as follows:
$$S_{ij} = \min\left(u_{ik}, u_{jk}\right), k = 1, 2, ..., n \tag{5}$$

That is, the minimum value of the membership degree of samples belonging to these two categories.
When the division of the two categories is relatively clear, it indicates that the membership degree of
samples belonging to a certain category must be greater than other values. Therefore, the better the
clustering result is, the smaller Sij should be. And the total dispersion is defined as:





$$S = \max_{i=1, j=1, i \neq j}^{c} S_{ij}$$
(6)

The smaller the dispersion is, the greater the difference between the two classes is and the better the
clustering result is.
Based on this, the clustering effectiveness index Vcs is defined as follows:
$$V_{cs} = \frac{C}{S}$$
(7)

In conclusion, when C is larger and S value is smaller, Vcs is larger and the clustering effect is better.

(4)Calculating the value function J.

$$J = \sum_{j=1}^{N} \sum_{i=1}^{C} u_{ij}^{m} d^2 \left( X_j, V_i \right)$$
(8)

where N is the total number of observations, and j is the fuzzy objective function; d2 is the Euclidean
distance between the ith clustering center and the jth data point (Wang, 2008);
The operation is stopped when J is less than a certain threshold.

(5)Calculating the new matrix U and return to step 2

$$u_{ij} = \frac{1}{\sum_{k=1}^{C} (\frac{d_{ij}}{d_{kj}})^{2/(m-1)}}$$
(9)

Initialize C、M
and J

Calculating matrix U

Calculating clustering centers
$C_i$

NO — Whether J is qualified

YES

Output clustering centers

**Fig.6** A flowchart of FCM
*3.2 Factor analysis*
FA is a multivariate statistical analysis method, which studies the internal dependence of variables and
reduces some variables with intricate relations to a few comprehensive factors (Li et al., 2016). FA is





the inferred decomposition of observed data into two matrices. One matrix represents a set of underlying unobserved characteristics of the subject which giverise to the observed characteristics and the other explains the relationship between the unobserved and observed characteristics (Max R 2018). And the mathematical formula can be expressed as follow:

$$\sum_{i=1}^{C} \mu_{ij} = 1, j = 1,2......, n$$

(10)

Where X (x1,x2,....,xp) is the original factor,F (F1,F2,...,Fm) is the common factor ;A= (akj) p×m is factor load matrix,akj represents the load of the K original factor on the J common factor; ε =(1,ε2,…,εp )is a special factor.

The main calculation steps of factor analysis method can be divided into six steps:

Test the feasibility of FA of original evaluation index variables

In this paper, SPSS was used to provide Bartlett sphericity test to determine whether variables are suitable for FA.

Standardized calculation of original data

In order to eliminate the numerical differences of different variables in order of magnitude and dimension, the original data should be standardized. And this paper adopted the Z standardization method in SPSS software.

Construct a common factor F

In the study, the first m factors for which the cumulative variance contribution rate is no less than 85%,were selected as common factors to represent the original data.

Factor rotation

In this paper, varimax orthogonal rotation was used to realize factor rotation.

Calculating factor scores;

The most common method for calculating factor scores is the Thomson regression method (Max R 2018), and the formula is as follow:

$$F = A^{'}R^{-1}X$$

(11)

where $A^{'}R^{-1}$ is factor scoring coefficient matrix and A is the factor loading matrix after rotation.

Calculating weight

The product of factor score coefficient and variance contribution rate is the contribution of each factor in the sample, and the sum of the contribution of each factor divided by the contribution of all indexes is the weight of each factor. It is expressed by the formula:

$$\omega_i = \frac{\sum_{j=1}^{m} \beta_{ji} e_j}{\sum_{i=1}^{p} \sum_{j=1}^{m} \beta_{ji} e_j}$$

(12)

where i=1,2,...,p;j=1,2,...,m; e is the contribution rate of factor variance.


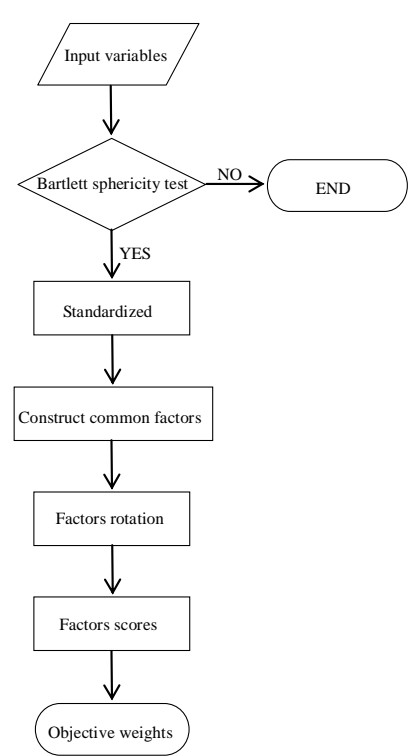

**Fig.7** A flowchart of FA

### *3.3 Combination weighting method*

Considering the defects of the current method for determining the weight of factors, the combination of
analytic hierarchy process and factor analysis method is used to determine the weight of each
influencing factor of debris flow.

### *3.3.1 Analytic hierarchy process (AHP)*

Analytic hierarchy process (AHP) was first proposed by Saaty (1979), a famous American
mathematician. It decomposes the factors related to decision-making into multiple layers, such as target
layer, criterion layer and scheme layer. AHP is a subjective weighting method and has obvious
advantages in determining the weight of each factor. The specific steps are as follows:
1 Establishing hierarchical structure model
The hierarchical structure is mainly divided into three layers: target layer, criterion layer and
scheme layer.
2 Establishing the judgment matrix
For the same level, judgment matrix is established by pair-wise comparison. The formula is as
follow:




$$A = \left(a_{ij}\right)_{n \times n}, a_{ij} > 0, a_{ij} = \frac{1}{a_{ji}}, \left(i, j = 1, 2, ...n\right)$$

(13)


where bij is the ratio of relative importance between element Bi and Bj, which is usually expressed by
the scoring method from 1 to 9(Saaty, 1977), as shown in table 2.
3 Consistency testing

The consistency test is divided into three steps:


(1) Calculate the consistency index(CI)(Saaty, 1977)and the expression is:


$$CI = \frac{\lambda_{max} - n}{n - 1}$$

(14)


(2) Average random consistency RI;

RI is associated with the order of judgment matrix, and their relationship is shown in Table 3.


(3) Obtaining the test coefficient CR.


$$CR = \frac{CI}{RI}$$

(15)


If CR<0.1, judgment matrix has a good consistency with reasonable judgment. Otherwise, the
judgment matrix needs to be revised until the consistency test is satisfied.

**Table 1** The random average consistency index

| n | 1 | 2 | 3 | 4 | 5 | 6 | 7 | 8 | 9 | 10 | 11 | 12 |
|---|---|---|---|---|---|---|---|---|---|----|----|----|
| RI | 0 | 0 | 0.52 | 0.89 | 1.12 | 1.26 | 1.36 | 1.41 | 1.46 | 1.49 | 1.52 | 1.54 |


**Table 2** Definition of comparative importance

| 1 | Two decision factors (e.g., indicators) are equally important |
|---|---|
| 3 | One decision factor is more important |
| 5 | One decision factor is strongly more importan |
| 7 | One decision factor is very strongly more important |
| 9 | One decision factor is extremely more important |
| 2,4,6,8 | Intermediate values |
| Reciprocals | If a ij is the judgment value when i is compared to j. Then U_ji = 1/U_ij is the judgment value when j is compared to i |

*3.3.2 Combination weighting rule*
The weight value obtained by AHP is set as ωci, and the weight value obtained by FA is set as ωyi
(Feng et al., 2010), as shown in Eq16.





$$\begin{cases} Min = \sum_{i=1}^{m}\sum_{j=1}^{n}\left(\alpha r_{ij}\omega^{c}{}_{i} - \beta r_{ij}\omega^{y}{}_{i}\right) \\ \alpha + \beta = 1 \end{cases}$$

(16)

Where α and β are weight coefficients calculated through AHP and factor analysis method. And α and β
are determined according to the following formula:
$$\begin{cases} \alpha = \sum_{i=1}^{m}\sum_{j=1}^{n} r_{ij}{}^{2}\omega^{y}{}_{i}\left(\omega^{c}{}_{i} + \omega^{y}{}_{i}\right) / \sum_{i=1}^{m}\sum_{j=1}^{n} r_{ij}{}^{2}\left(\omega^{c}{}_{i} + \omega^{y}{}_{i}\right)^{2} \\ \beta = \sum_{i=1}^{m}\sum_{j=1}^{n} r_{ij}{}^{2}\omega^{c}{}_{i}\left(\omega^{c}{}_{i} + \omega^{y}{}_{i}\right) / \sum_{i=1}^{m}\sum_{j=1}^{n} r_{ij}{}^{2}\left(\omega^{c}{}_{i} + \omega^{y}{}_{i}\right)^{2} \end{cases}$$

(17)

And the combined weight ($\omega^{z}{}_{i}$) can be represented in Eq18:
$$\omega_{i}^{z} = \alpha\omega_{i}^{c} + \beta\omega_{i}^{y}$$

(18)

## 3.4 Efficiency coefficient method

Based on the principle of multi-objective programming, the efficiency coefficient method transforms
each factor into a measurable evaluation score through the efficiency function, and combines the
weight of factors to make a comprehensive evaluation. The specific steps are as follows:
1 Selecting evaluation factors
2 Determine the satisfactory value and the unallowable value
The satisfaction value is a value based on years of experience, while the unallowable value is the
lowest or highest acceptable value of the evaluation index.
3 Calculating the single efficacy coefficient
The single efficacy coefficient was calculated by the corresponding efficacy function based on the
sensitivity of each factor. And It is mainly divided into three variables: the extremely large variable (the
higher the factor, the higher the efficiency coefficient), the infinitesimal variable (the smaller the index
value, the larger the efficiency coefficient value) and the Interval variable (The value reach the highest
in a certain interval). The specific formula is as follows:
$$g_{1i} = \begin{cases} \dfrac{x_i - x_{ni}}{x_{yi} - x_{ni}} \times 40 + 60, x_i < x_{yi} \\ 100, x_i \geq x_{yi} \end{cases}$$

(19)

where $g_{1i}$ is the single efficacy coefficient value of the $i^{th}$ extremely large factor; $X_i$ is the actual value
of the $i^{th}$ factor; $X_{yi}$ is the satisfactory value of the $i^{th}$ factor; $X_{ni}$ is the unallowable value of the $i^{th}$
factor.
The infinitesimal variable:
$$g_{2i} = \begin{cases} \dfrac{x_i - x_{ni}}{x_{yi} - x_{ni}} \times 40 + 60, x_i > x_{yi} \\ 100, x_i \geq x_{yi} \end{cases}$$

(20)

The Interval variable:



$$g_{3i} = \begin{cases} \left(1 - \dfrac{x_{\min} - x_i}{x_{\min} - x_{n\min}}\right) \times 40 + 60, \, x_i < x_{\min} \\ 100, \, x_{\min} < x_i < x_{\max} \\ \left(1 - \dfrac{x_i - x_{\max}}{x_{n\max} - x_{\max}}\right) \times 40 + 60, \, x_i > x_{\min} \end{cases}$$

(21)


4 Calculating the total efficiency coefficient

$$G = \sum_{i}^{m} \left(g_i \omega_i\right)$$

(22)

where G is the total efficacy coefficient, $g_i$ is the single efficacy coefficient and $\omega_i$ is the weight  of the
$i^{th}$ factor.
The flow chart for the method used for our classification and susceptibility analysis is shown in
Fig. 6.

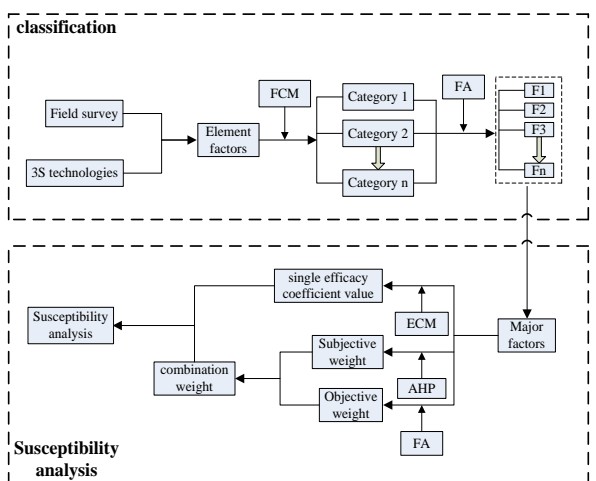


**Fig.8** Flow chart used for classification and susceptibility where category 1 to category n represent clustering results and F1 to Fn represent the major factors selected by FA.

*3.5 Influencing Factors*
The topographical, geological and climatic factors play a critical role in the distribution and activities
of debris flows (B. F. DI et al., 2008). Table 2 shows the influencing factors selected by researches in
debris flow susceptibility assessment in recent years. Rainfall is one of the most pivotal external factors
inducing debris flow disasters, but the meteorological data in our area are all from the same station,
which cannot reflect the differences between each catchment. Therefore, rainfall was not included in
this study. In addition, the frequency of debris flow and the size of soil particles are difficult to obtain
accurately. The loose material volume reflects the lithological characteristics and fault length to some
extent, so lithology and fault length were not taken into account. The basin area, main channel length,



drainage density, average slop angle, average gradient of main channel, vegetation coverage, maximum
elevation difference and curvature of the main channel, which were cited and available, were selected
in this paper. As source conditions, the loose material volume and the loose material supply length ratio
were also considered. As the study area is located in a tourist area with a relatively dense population,
population density is selected as the factor of human activities. A total 13 influencing factors were
selected based on the previous research findings to reflect the characteristics of the watershed. All these
factors were acquired in our field survey or calculated in ArcGIS, as described below.
**Table 3** Factors frequently used in susceptibility ananlysis of debris flow

| Factors | Lin (2002) | Chang (2006) | Chang (2007) | Lu (2007) | Meng (2010) | Zhang (2011) | Zhang (2013) | Shi (2016) | Niu (2014) | Time |
|---|---|---|---|---|---|---|---|---|---|---|
| Rainfall intensity | | √ | √ | | | | | | | 2 |
| Daily rainfall | | | √ | | | √ | √ | | | 3 |
| Cumulative rainfall | | √ | √ | | | | | | | 2 |
| Main channel length | | √ | √ | | √ | √ | | √ | √ | 6 |
| Average slope angle | √ | √ | √ | | √ | | √ | √ | √ | 7 |
| Drainage density | | √ | √ | | √ | √ | √ | √ | √ | 7 |
| Soil particle size | | √ | √ | | | | | | | 2 |
| Basin area | √ | √ | √ | √ | √ | √ | | √ | √ | 8 |
| Average gradient of main channel | √ | | √ | | | √ | √ | √ | √ | 6 |
| Frequency | √ | | | | √ | √ | | | | 3 |
| Loose material volume | | | | | √ | | √ | √ | √ | 4 |
| Vegetation coverage | √ | | | √ | √ | | √ | | √ | 5 |
| Population density | | | | | | √ | | | | 1 |
| Lithology | √ | | | | | | | | √ | 2 |
| Maximum elevation difference | | | | | √ | √ | | √ | √ | 4 |
| Curvature of the main channel | | | | | | √ | √ | √ | √ | 4 |
| Fault length | √ | | | | | | | | | 1 |



Basin area(F1)(km$^2$)

Basin area reflects the scale of debris flow. Generally, the larger the basin area is, the greater the

risk of debris flow will be. It was obtained by geometric operations in ArcGIS and corrected by the
remote sensing image in Google earth.
Main channel length(F2)(km)

Main channel length reflects the potential for increasing loose sources along the route. This value

was measured from ArcGIS by combining RS technology and topographic map.
Drainage density(F3)(km/km$^2$)

Drainage density is the ratio of the total drainage length to the watershed area and it is an

important index to describe the degree of ground being cut by gullies.
Average gradient of main channel(F4)

It is the ratio of the maximum elevation difference of main channel to its linear length. The larger

the value, the better the hydrodynamic condition is. This value is obtained from the DEM.
Average slop angle(F5)(°)

As F5 increases, the erosion capacity and intensity of precipitation increase. The value was

obtained by ArcGIS slope analysis tool.
Maximum elevation difference(F6)(m)

The difference between the maximum and minimum elevation values in the basin provides kinetic

energy condition of disaster. This value is also obtained from the DEM.
Curvature of the main channel(F7)

F7 is the ratio of the main channel length to its linear length, which reflects the degree of channel

blockage.
The loose material volume (F8)(×104m$^3$)

The loose material is one of fundamental factors triggering debris flows. This factor is obtained

through field investigation with tape and laser rangefinder. And the thickness was obtained by field
estimation and trench test.
The loose material supply length ratio (F9)

F9 is the ratio of loose material length along a channel to total channel length, which reflects the

successive supplied sediments. It was obtained through field survey and RS technology.
Vegetation coverage(F10)

The lower the vegetation coverage will be, the more serious the soil erosion. It was estimated from

field survey and SPOT5 imaging.
Population density(F11)(quantity/km$^2$)

With the development of social economy, human activities have gradually become an important

factor affecting debris flow. Population density reflects the intensity of human activities, which is
estimated according to the number of buildings through field survey and RS technology.
Roundness(F12)

Roundness is the morphological statistical element of gully, and the plane shape of gully variates

from its developmental stage. F12 is the ratio of the length of main channel of debris flow to its area.
The most volume of once flow(F13)(×104m$^3$)

Liu (1993) selected F13 as the main factor in the risk assessment of debris flow, which is one of

the important factors to evaluate the degree of debris flow hazard.
**Table 4** The values for the 13 factors of the 21 debris flow catchments

| | F1 | F2 | F3 | F4 | F5 | F6 | F7 | F8 | F9 | F10 | F11 | F12 | F13 |
|---|---|---|---|---|---|---|---|---|---|---|---|---|---|

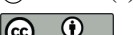


| | | | | | | | | | | | | | |
|---|---|---|---|---|---|---|---|---|---|---|---|---|---|
| 1 | 1.887 | 1.721 | 2.51 | 0.48 | 25.88 | 639 | 1.09 | 1.04 | 0.36 | 0.5 | 28 | 0.74 | 2.41 |
| 2 | 0.907 | 0.984 | 1.85 | 0.7 | 26.77 | 579 | 1.09 | 0.706 | 0.65 | 0.5 | 8 | 0.68 | 0.49 |
| 3 | 0.292 | 0.321 | 1.71 | 0.24 | 25.27 | 371 | 1.22 | 0.167 | 0.33 | 0.45 | 90 | 0.82 | 0.05 |
| 4 | 2.057 | 2.296 | 2.05 | 0.44 | 27.17 | 752 | 1.1 | 1.615 | 0.74 | 0.55 | 27 | 0.62 | 2.33 |
| 5 | 1.547 | 1.728 | 1.6 | 0.42 | 25.44 | 610 | 1.18 | 0.956 | 0.41 | 0.45 | 25 | 0.58 | 1.62 |
| 6 | 2.77 | 3.113 | 2.16 | 0.32 | 25 | 745 | 1.15 | 1.616 | 0.77 | 0.65 | 6 | 0.61 | 5.95 |
| 7 | 1.223 | 1.098 | 1.96 | 0.58 | 23.51 | 584 | 1.12 | 0.7 | 0.61 | 0.6 | 9 | 0.77 | 0.66 |
| 8 | 0.445 | 0.898 | 2.07 | 0.49 | 19.8 | 386 | 1.19 | 0.463 | 0.69 | 0.65 | 23 | 0.66 | 0.18 |
| 9 | 0.34 | 0.396 | 1.25 | 1.06 | 25.81 | 381 | 1.12 | 0.29 | 0.73 | 0.6 | 16 | 0.71 | 0.06 |
| 10 | 6.65 | 3.539 | 1.98 | 0.27 | 22.46 | 856 | 1.08 | 18.457 | 0.48 | 0.52 | 102 | 0.68 | 5.04 |
| 11 | 0.388 | 0.965 | 2.57 | 0.37 | 22.56 | 508 | 1.11 | 0.397 | 0.75 | 0.55 | 105 | 0.43 | 0.19 |
| 12 | 0.713 | 0.787 | 2.74 | 0.63 | 22.35 | 366 | 1.16 | 0.564 | 0.62 | 0.55 | 145 | 0.72 | 0.21 |
| 13 | 6.319 | 4.539 | 2.13 | 0.22 | 22.89 | 828 | 1.12 | 5.549 | 0.35 | 0.6 | 22 | 0.6 | 6.75 |
| 14 | 0.664 | 1.036 | 1.61 | 0.54 | 25.31 | 550 | 1.13 | 0.956 | 0.66 | 0.7 | 62 | 0.48 | 0.29 |
| 15 | 0.492 | 0.51 | 1.3 | 0.77 | 25.66 | 368 | 1.09 | 0.13 | 0.68 | 0.6 | 230 | 0.71 | 0.07 |
| 16 | 1.093 | 1.564 | 1.95 | 0.41 | 24.55 | 568 | 1.22 | 1.027 | 0.72 | 0.65 | 30 | 0.59 | 0.75 |
| 17 | 5.312 | 4.564 | 1.55 | 0.18 | 24.78 | 743 | 1.03 | 6.443 | 0.31 | 0.62 | 14 | 0.43 | 4.04 |
| 18 | 0.85 | 1.289 | 2.04 | 0.47 | 20.99 | 571 | 1.07 | 1.196 | 0.74 | 0.6 | 120 | 0.53 | 0.6 |
| 19 | 0.425 | 0.901 | 2.17 | 0.56 | 22.49 | 479 | 1.09 | 0.451 | 0.62 | 0.55 | 165 | 0.66 | 0.22 |
| 20 | 1.71 | 2.334 | 1.77 | 0.26 | 17.27 | 583 | 1.05 | 1.313 | 0.71 | 0.55 | 182 | 0.5 | 3.59 |
| 21 | 3.804 | 3.32 | 1.57 | 0.25 | 18.46 | 668 | 1.2 | 0.4317 | 0.58 | 0.65 | 66 | 0.49 | 6.31 |

## 4 Result

### 4.1 Fuzzy c-means clustering analysis

The curve of clustering effectiveness index Vcs with the number of clustering centers is shown in Fig. 9 and the optimal number of clustering of evaluation units is 4. Based on the basic data of 21 debris flows, the FCM was carried out and set the fuzzy weighted index m=2. And results were shown in table 5.




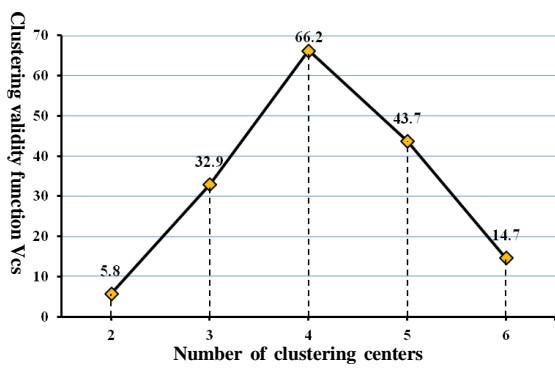

**Fig.9** Clustering validity function Vcs

| Category | Catchment |
|---|---|
| I | 1、2、5、7、14、16、21 |
| II | 4、6、10、13、17、 |
| III | 11、18、19、20 |
| IV | 3、8、9、12、15 |

**Table 5** Clustering results of 21 debris flows debris flows

Thus 21 debris flows in the study area are divided into 4 categories. The data of each catchment belonging to the same category have certain internal similarity and vary greatly among different categories. In other words, data of different influencing factors have different effects on different types of debris flows, which provide a favorable basis for us to analyze the main influencing factors of debris flows, and also points out the direction for monitoring and prevention of debris flows.

## 4.2 Factor analysis

Based on the clustering results of 21 debris flows, FA was used to analyze each type of debris flow. Table 2, table 3, table 4 and table 5 are the results of the first, second, third and fourth categories, respectively.

As shown in table 2, in the first category, the accumulative contribution rate of the first three factors (C1, C2 and C3) reaches 86.40%, which retain most information of the 13 original variables. For the first group, the load values of the main factors 1, 2 and 3 are relatively large in the basin area, the most volume of once flow, the maximum elevation difference, the main channel length and curvature of the main channel, population density and drainage density, respectively. Similarly, in the second type, the load values of the main factors 1, 2 and 3 are relatively large in the basin area, the main channel length and population density, loose material volume and drainage density, maximum elevation difference, respectively. In the third category, the load values of the main factors 1, 2 and 3 are relatively large in the basin area, main channel length, the most volume of once flow, loose material volume and the loose material supply length ratio and vegetation coverage, respectively. And In the fourth category, the load values of the main factors 1, 2 and 3 are relatively large in main channel length, drainage density, loose material volume, the most volume of once flow and the loose material supply length ratio and population density, respectively.

Among the 13 factors, the basin area and the most volume of once flow reflect the scale of debris flow eruption. The main channel length, drainage density, average gradient of main channel, the





average slope, maximum elevation difference, curvature of the main channel, roundness reflect the
topographical condition. The loose material volume and the loose material supply length ratio are the
material sources for debris flow. Vegetation coverage reflects geomorphologic condition. Population
density reflects the impact of human activities on nature to some extent. Therefore, four types of debris
flows can be named according to the results of FCM and FA.
The first category can be defined as debris flow closely related to scale-topography-human
activities. Considering the situation, monitoring and control of basic material sources is recommended.
Similarly, the second, third, and fourth categories can be defined as topography-human
activities-provenance, scale-provenance-topography topography-scale-provenance-human activities,
respectively. In the same way, corresponding prevention measures can be proposed according to the
characteristics of each type of debris flow.
**Table 6** The factor load matrix after rotation and contribution ratios for the first category

| Factor | C1 | C2 | C3 |
|--------|------|------|------|
| F1 | 0.960 | 0.258 | 0.094 |
| F2 | 0.876 | 0.46 | 0.092 |
| F3 | -0.101 | -0.465 | 0.589 |
| F4 | -0.611 | -0.739 | -0.17 |
| F5 | -0.832 | -0.356 | 0.349 |
| F6 | 0.902 | 0.053 | 0.422 |
| F7 | 0.239 | 0.737 | -0.164 |
| F8 | -0.776 | 0.2 | 0.569 |
| F9 | -0.272 | 0.102 | -0.891 |
| F10 | -0.017 | 0.492 | -0.683 |
| F11 | 0.306 | 0.798 | -0.193 |
| F12 | -0.077 | -0.869 | 0.316 |
| F13 | 0.938 | 0.311 | 0.084 |
| Contribution rate (%) | 51.686 | 24.245 | 10.469 |
| Accumulative contribution (%) | 51.686 | 75.931 | 86.399 |

**Table 7** The factor load matrix after rotation and contribution ratios for the second category

| Factor | C1 | C2 | C3 |
|--------|------|------|------|
| F1 | 0.850 | 0.497 | -0.154 |
| F2 | 0.937 | -0.130 | -0.301 |
| F3 | -0.203 | 0.090 | 0.961 |
| F4 | -0.944 | 0.073 | 0.303 |
| F5 | -0.853 | -0.467 | -0.208 |
| F6 | 0.485 | 0.801 | 0.301 |
| F7 | -0.103 | -0.230 | 0.968 |
| F8 | 0.389 | 0.869 | -0.148 |
| F9 | -0.808 | -0.143 | 0.500 |
| F10 | 0.280 | -0.925 | 0.108 |
| F11 | 0.075 | 0.980 | -0.002 |



| Factor | | | |
|---|---|---|---|
| F12 | -0.247 | 0.632 | 0.735 |
| F13 | 0.690 | -0.105 | 0.595 |
| Contribution rate (%) | 45.350 | 31.221 | 20.737 |
| Accumulative contribution (%) | 45.350 | 76.572 | 97.309 |

**Table 8** The factor load matrix after rotation and contribution ratios for the third category

| Factor | C1 | C2 | C3 |
|---|---|---|---|
| F1 | 0.986 | 0.161 | -0.043 |
| F2 | 0.966 | 0.218 | -0.136 |
| F3 | -0.931 | 0.318 | -0.181 |
| F4 | -0.590 | -0.739 | 0.325 |
| F5 | -0.981 | -0.171 | 0.094 |
| F6 | 0.806 | 0.415 | 0.423 |
| F7 | -0.965 | 0.128 | -0.230 |
| F8 | 0.882 | 0.142 | 0.450 |
| F9 | 0.044 | 0.938 | 0.343 |
| F10 | 0.042 | 0.054 | 0.998 |
| F11 | 0.705 | -0.571 | -0.421 |
| F12 | -0.044 | -0.996 | 0.075 |
| F13 | 0.949 | 0.160 | -0.273 |
| Contribution rate (%) | 61.553 | 24.036 | 14.411 |
| Accumulative contribution (%) | 61.553 | 85.589 | 100 |

**Table 9** The factor load matrix after rotation and contribution ratios for the fourth category

| Factor | C1 | C2 | C3 |
|---|---|---|---|
| F1 | 0.749 | 0.239 | 0.610 |
| F2 | 0.937 | 0.258 | -0.110 |
| F3 | 0.913 | -0.314 | 0.184 |
| F4 | -0.249 | 0.875 | 0.068 |
| F5 | -0.900 | -0.002 | 0.374 |
| F6 | 0.051 | 0.293 | -0.953 |
| F7 | 0.328 | -0.840 | -0.431 |
| F8 | 0.918 | 0.105 | -0.123 |
| F9 | 0.216 | 0.971 | -0.093 |
| F10 | 0.302 | 0.873 | -0.305 |
| F11 | -0.068 | 0.053 | 0.919 |
| F12 | -0.455 | -0.844 | 0.219 |
| F13 | 0.994 | 0.090 | 0.037 |
| Contribution rate (%) | 44.768 | 30.086 | 19.917 |
| Accumulative contribution (%) | 44.768 | 74.854 | 94.771 |




### 4.3 Weights of major factors


Based on FA of each category of debris flow in the previous section, the main influencing factors were
obtained. However, the repeatability of evaluation information should be reduced. Average slop angle
and average gradient of main channel are both indicators of potential energy, so the average gradient of
main channel is omitted. Similarly, curvature of the main channel, the loose material supply length
ratio and roundness were omitted. So 9 factors, including basin area F1, main channel length F2,
drainage density F3, average slop angle F5, maximum elevation difference F6, the loose material
volume F8, vegetation coverage F10, population density F11 and the most volume of once flow F13
were selected. On the other hand, a reduction in the number of indicators facilitates the allocation of
weight values.

### 4.3.1 Subjective weights


Analytic hierarchy process(AHP)was applied to calculate the subjective weight in this paper. The
hierarchical structure (Fig. 10) was constructed, and the 1-9 scale method was used to grade each factor.
The judgment matrices A-A ' (Table 10) and B-B' (Table 11) were constructed and the consistency test
was conducted, respectively. The weight values of each factor are shown in table 12.

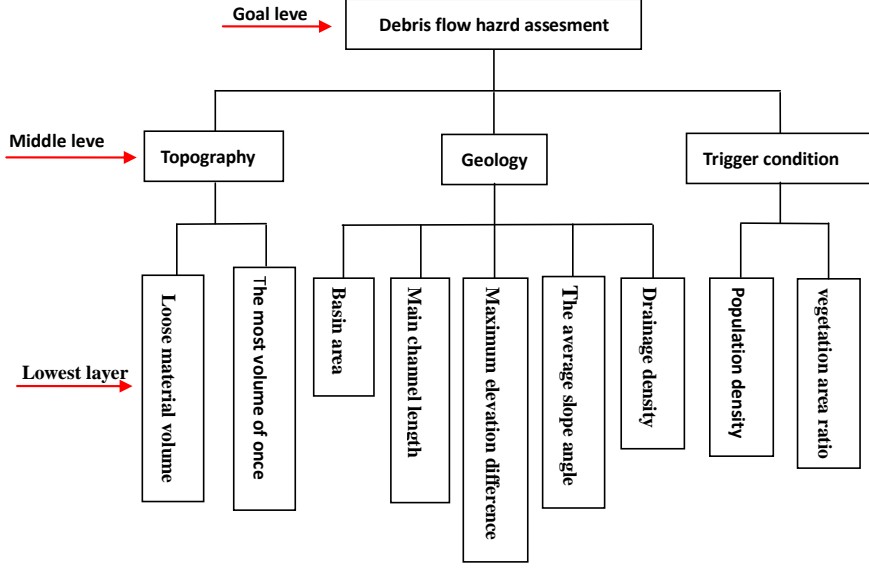


**Fig.10** Hierarchical structure for debris flow susceptibility analysis
**Table 10** Comparison matrix elements for geology condition

| Geology | F1 | F2 | F6 | F5 | F3 | CI | RI | CR |
|---------|------|------|------|------|------|----|----|----|
| F1 | 1.00 | 2.00 | 2.00 | 3.00 | 3.00 | | | |
| F2 | 0.50 | 1.00 | 1.00 | 3.00 | 3.00 | | | |
| F6 | 0.50 | 1.00 | 1.00 | 3.00 | 3.00 | | | |





| | | | | | | | | |
|---|---|---|---|---|---|---|---|---|
| F5 | 0.33 | 0.33 | 0.33 | 1.00 | 1.00 | | | |
| F3 | 0.33 | 0.33 | 0.33 | 1.00 | 1.00 | 0.0024 | 0.52 | 0.0045 |

CR=0.0045<0.1, met the conformance inspection requirements.

**Table 11** Comparison matrix elements of the criterion level factors

| **middle leve** | Topography | Geology | Trigger condition | CI | RI | CR |
|---|---|---|---|---|---|---|
| Topography | 1.00 | 1.50 | 2.00 | | | |
| Geology | 0.67 | 1.00 | 1.50 | | | |
| Trigger condition | 0.50 | 0.67 | 1.00 | 0.02 | 1.12 | 0.02 |

CR=0.02<0.1, met the conformance inspection requirements.

**Table 12** The weighted values of the factors obtained by AHP

| **Factor** | F1 | F2 | F3 | F5 | F6 | F8 | F10 | F11 | F13 |
|---|---|---|---|---|---|---|---|---|---|
| **Weight** | 0.11 | 0.07 | 0.03 | 0.03 | 0.07 | 0.28 | 0.06 | 0.17 | 0.18 |

### 4.3.2 Objective weights

FA was applied to calculate the objective weight in this paper. The weight values of each factor are shown in table 13.

**Table 13** The weighted values of the factors obtained by factor analysis

| **Factor** | F1 | F2 | F3 | F5 | F6 | F8 | F10 | F11 | F13 |
|---|---|---|---|---|---|---|---|---|---|
| **Weight** | 0.15 | 0.17 | 0.05 | 0.01 | 0.17 | 0.08 | 0.07 | 0.14 | 0.16 |

### 4.3.3 Combination weights

After the subjective weight and objective weight are obtained, the respective distribution coefficients are solved according to eq1 and the final combined weight values of each factor are shown in table 14, α=0.70,β=0.30, F8>F13>F11>F1>F2=F6>F10>F3>F5.

**Table 14** The combined weighted values of the factors

| **Factor** | F1 | F2 | F3 | F5 | F6 | F8 | F10 | F11 | F13 |
|---|---|---|---|---|---|---|---|---|---|
| **Combination Weight** | 0.12 | 0.10 | 0.04 | 0.03 | 0.10 | 0.22 | 0.06 | 0.16 | 0.17 |

### 4.4 The efficacy coefficient of factors

Among the 9 factors, basin area, main channel length, drainage density, maximum elevation difference, the loose material volume, the most volume of once flow and population density are all extremely large variables. Vegetation coverage is the infinitesimal variable. And Average slop angle is an interval variable. Table 15 shows the efficacy coefficient scores of 21 debris flows after combined with weight calculation.

**Table 15** The efficacy coefficient scores of 21 debris flows





|    | F1    | F2    | F3    | F5    | F6    | F8   | F10  | F11   | F13  | Total score |
|----|-------|-------|-------|-------|-------|------|------|-------|------|-------------|
| 1  | 13.56 | 12.89 | 8.53  | 7.30  | 8.29  | 2.26 | 3.57 | 10.29 | 5.84 | 72.53       |
| 2  | 13.40 | 10.90 | 7.78  | 6.61  | 7.79  | 2.26 | 2.90 | 9.72  | 5.84 | 67.19       |
| 3  | 13.14 | 10.44 | 7.31  | 5.98  | 6.08  | 2.26 | 2.76 | 12.08 | 6.35 | 66.39       |
| 4  | 13.83 | 12.81 | 8.66  | 7.84  | 9.22  | 2.26 | 3.10 | 10.26 | 5.33 | 73.31       |
| 5  | 13.52 | 12.07 | 8.27  | 7.31  | 8.05  | 2.26 | 2.65 | 10.21 | 6.35 | 70.67       |
| 6  | 13.83 | 16.57 | 9.20  | 8.61  | 9.16  | 2.06 | 3.21 | 9.66  | 4.32 | 76.62       |
| 7  | 13.40 | 11.07 | 8.02  | 6.71  | 7.83  | 1.92 | 3.01 | 9.75  | 4.82 | 66.54       |
| 8  | 13.28 | 10.57 | 7.42  | 6.53  | 6.21  | 1.59 | 3.12 | 10.15 | 4.32 | 63.19       |
| 9  | 13.20 | 10.45 | 7.34  | 6.05  | 6.17  | 2.26 | 2.28 | 9.95  | 4.82 | 62.52       |
| 10 | 21.88 | 15.62 | 12.18 | 9.01  | 10.07 | 1.83 | 3.03 | 12.42 | 5.64 | 91.67       |
| 11 | 13.25 | 10.58 | 7.38  | 6.59  | 7.21  | 1.84 | 3.64 | 12.51 | 5.33 | 68.33       |
| 12 | 13.33 | 10.60 | 7.63  | 6.42  | 6.04  | 1.82 | 3.80 | 13.66 | 5.33 | 68.64       |
| 13 | 15.71 | 17.40 | 11.92 | 9.95  | 9.84  | 1.87 | 3.18 | 10.12 | 4.82 | 84.82       |
| 14 | 13.52 | 10.69 | 7.59  | 6.66  | 7.56  | 2.26 | 2.65 | 11.27 | 3.81 | 65.99       |
| 15 | 13.13 | 10.46 | 7.46  | 6.16  | 6.06  | 2.26 | 2.34 | 16.10 | 4.82 | 68.78       |
| 16 | 13.55 | 11.17 | 7.92  | 7.15  | 7.70  | 2.02 | 3.00 | 10.35 | 4.32 | 67.18       |
| 17 | 16.14 | 14.58 | 11.15 | 9.97  | 9.14  | 2.04 | 2.59 | 9.89  | 4.62 | 80.13       |
| 18 | 13.63 | 11.01 | 7.73  | 6.89  | 7.73  | 1.69 | 3.09 | 12.94 | 4.82 | 69.54       |
| 19 | 13.28 | 10.61 | 7.41  | 6.53  | 6.97  | 1.83 | 3.22 | 14.23 | 5.33 | 69.42       |
| 20 | 13.69 | 14.12 | 8.39  | 7.88  | 7.83  | 1.35 | 2.82 | 14.72 | 5.33 | 76.12       |
| 21 | 4.60  | 15.57 | 12.25 | 14.93 | 14.59 | 0.34 | 3.89 | 9.90  | 4.87 | 80.94       |

## *4.5 Susceptibility assessment of debris flow*

Taking the total efficiency coefficient of each catchment as the evaluation standard (the larger the value
is, the higher the possibility of debris flow), FCM was conducted for 21debris flow in the study area.
The result showed that the susceptibility of debris flow was divided into three grades: high (H),
moderate (m) and low (L). Combined with the classification of each debris flow mentioned above, the
final results were shown in the table 16.

| Catchment | Category | Susceptibilit level |
|-----------|----------|---------------------|
| 1         | I        | M                   |
| 2         | I        | L                   |
| 3         | IV       | L                   |
| 4         | II       | M                   |
| 5         | I        | L                   |
| 6         | II       | M                   |
| 7         | I        | L                   |
| 8         | IV       | L                   |
| 9         | IV       | L                   |

**Table 16** The qualitative description and susceptibility class for each debris flow catchment




| 10 | II | H |
|---|---|---|
| 11 | III | L |
| 12 | IV | L |
| 13 | II | H |
| 14 | I | L |
| 15 | IV | L |
| 16 | I | L |
| 17 | II | M |
| 18 | III | L |
| 19 | III | L |
| 20 | III | M |
| 21 | I | M |

As shown in table 16, susceptibility for the 10th and 13th catchments was high and both of
them belong to the debris flow with close relationship between topography, human activities and
provenance. Susceptibility for 6 catchments, including the 1st, 4th, 6th, 17th, 20th and 21th, had
medium susceptibility. The other 13 had low susceptibility.
Normative scoring, k-means clustering algorithm and hierarchical cluster were determined to
validate susceptibility analysis methods used in this paper.
Based on the field investigation, the 10th catchment is located in Huangsongyu national
Mining Park, where a large amount of slag has been accumulated. With low vegetation coverage
and steep terrain, the gully was in its prime, which directly threatened the safety of villagers and
tourists. What's more, there are several warning boards of natural disaster and corresponding
monitoring equipment in the scenic spot(as shown in Fig.5. And the 13th catchment is located
Lishugou village scenic spot. Part of the pedestrian passageway was built, but a lot of stones were
piled up in the trench and the road was broken and steep(as shown in Fig.6). However, there is no
obvious accumulation of loose materials in the catchments with low susceptibility. The gully was
in its old stage with high vegetation coverage and little human interference. The quantitative
comprehensive evaluation results of debris flow susceptibility are shown in table 17, which are
divided into two levels: low (L) and moderate (M). Among them , the susceptibility of the 10th
and the 13th catchments were moderate and the others were low.
The K-means algorithm (K) (1978) and Hierarchical cluster (H) (2017) were used for the
classification of our data to measure the classification performance in this paper. And the results
were shown in table 17. The susceptibility results obtained by K and FCM are exactly the same.
The susceptibility assessment of 17th and 21th were high based on H and moderate from FCM and
K. However, such minor differences are acceptable. On the other hand, the susceptibility results
obtained by FCM and normative scoring are different. This is mainly because the number of
categories is different and the level was generally higher obtained by FCM. In addition, it can be
seen from the tree graph(Fig.11)obtained by Hierarchical cluster, that the clustering results are
more reasonable to be divided into three categories, which is consistent with the Vcs. Therefore,
the susceptibility model established in this paper is suitable and reasonable.
**Table 17** Comparison of susceptibility analyses based on different algorithms

| | 1 | 2 | 3 | 4 | 5 | 6 | 7 | 8 | 9 | 10 | 11 | 12 | 13 | 14 | 15 | 16 | 17 | 18 | 19 | 20 | 21 |
|---|---|---|---|---|---|---|---|---|---|---|---|---|---|---|---|---|---|---|---|---|---|
| K | M | L | L | M | L | M | L | L | L | H | L | L | H | L | L | L | M | L | L | M | M |




| Hierarchical | M | L | L | M | L | M | L | L | L | H | L | L | H | L | L | L | H | L | L | M | H |
| --- | --- | --- | --- | --- | --- | --- | --- | --- | --- | --- | --- | --- | --- | --- | --- | --- | --- | --- | --- | --- | --- |
| FCM | M | L | L | M | L | M | L | L | L | H | L | L | H | L | L | L | M | L | L | M | M |


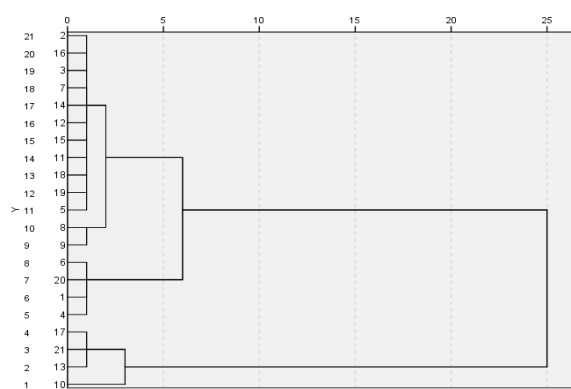

**Fig.11** Tree diagram obtained by Hierarchical cluster.


## 5 Discussion
The accuracy of the debris flow classification directly affects the development of prevention and
control measures. Based on different criteria, such as genetic classification, outbreak frequency,
material composition, the same debris flow can belong to multiple categories at the same time, which
does not reasonably reflect its multiple characteristics. In addition, the traditional classification
standard has some hysteresis to prevent debris flow. Considering that different types of debris flow
have different main influencing factors, the FCM and FA were combined in this study to refine and
summarize the importance of various factors to improve the accuracy of the classification. FCM is
different from traditional rigid division and it is based on the distance function to make the maximum
correlation between the same kind of data and the minimum correlation between different kinds of data.
The clustering effectiveness Vcs was introduced to effectively solve the problem of determining the
number of clusters, and the clustering analysis was carried out on the basic data of 21 debris flows. FA
is a primary exploratory tool for dimension reduction and visualization (Verde et al., 2018). The main
influencing factors of each category are obtained by FA, which not only realizes effective
dimensionality reduction but also eliminates the linear relationship between factors. The results showed
that different kinds of debris flows obtained by the FCM had different major influencing factors. In
other words, data for different influencing factors have different effects on different types of debris
flows, which demonstrate the advantages of the FCM when combined with the factor analysis.
According to different main influencing factors, the development characteristics of debris flows can be
reclassified. It also provided an effective basis for us to study the origin and classification of debris
flow and point out the direction for monitoring and controlling disasters.
The reasonable selection of evaluation factors is the premise of accurate evaluation of debris flow
susceptibility. In this paper, 13 factors were preliminarily selected based on previous experience and
field investigation conditions. And secondary screening was carried out based on FA analysis results,
which enhanced rationality of screening. The determination of the factor weight is crucial to accurately
evaluate the susceptibility of the debris flow. FA is a common objective evaluation method in statistical
analysis that determines the weight of factors according to the internal correlation and patterns of data.


However, the objective method cannot reflect the relative significance of each influencing factor and
may create misleading information. The AHP can make full use of expert experience and achievements
in the corresponding fields to evaluate the influencing factors, which is a subjective method. However,
different researchers have different preferences for major factors, which have a negative impact on the
results. Therefore, combination weighting, which combines the advantages of the FA and AHP, is
superior to the other methods alone when trying to obtain a more scientific and reasonable evaluation
result.
The efficiency coefficient method is different from other evaluation systems. By determining the
satisfaction value of each factor as the upper limit and the unallowable value as the lower limit, the
satisfaction degree is calculated through the corresponding efficiency function, and the final
comprehensive score was obtained based on the weight evaluation. This method not only considers the
relative importance of different factors but also determines the value based on the susceptibility to
debris flow. Therefore, the efficiency coefficient method can objectively evaluate complicated research
objects, such as debris flow, with this form of classification that conforms to people's logical thinking.
However, the evaluation method adopted in this paper also has limitations: (1) Fuzzy c-means
clustering is not applicable to the evaluation of a single debris flow gully; (2) Factor analysis method is
not applicable when the sample data is too small;(3) The tools used in field investigation are too simple
and some data, such as the loose material supply length ratio, are not accurate enough; (4) Rainfall
variations were not considered between different debris flow.

## 6 Conclusions

Classification and susceptibility analysis are of great significance for the early warning and prevention
of debris flow. Based on field investigation and "3S technology", an improved FCM and FA method
were used to establish classification model and obtain the main influencing factors of different types of
debris flow in the current study. And the ECM was used for the susceptibility analysis based on the
weights of major factors.
In this paper, 21 debris flows in Beijing were divided into 4 categories. Nine major factors
screened from the classification results were determined for susceptibility analysis using both the ECM
and combination weighting, and the susceptibility assessment was divided into 3 levels, which has been
validated with normative scoring, the K-means algorithm and hierarchical clustering. An effective
scientific classification and susceptibility assessment results of debris flow were obtained, which
provides a theoretical basis for formulating disaster prevention, reduction plans and measures for debris
flow. Therefore, a semi-quantitative evaluation method which combines fuzzy mathematics,
multivariate statistical analysis and geological environment, is suitable for risk assessment for a study
area with a limited number of samples. Different methods have their own advantages and
disadvantages, and some methods are complementary to a certain extent, so it is desirable to enhance
the rationality of the application through the combination of multiple methods.

## Data availability

The data used to support the findings of this study are included within the article.





## Author contribution:

Zhu Liang was responsible for the writing and graphic production of the manuscript. Changming Wang was responsible for the revision of the manuscript. Songling Han was responsible for the part of the calculation. Kaleem Ullah Jan Khan was responsible for the translation. Yiao Liu was responsible for the reference proofreading.

## Competing interests:

The authors declare that they have no conflict of interest.

## Acknowledgements

This work was supported by the National Natural Science Foundation of China (Grant No. 41572257).

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
