# Peer review of "Classification and susceptibility assessment of debris flow"

_Natural Hazards and Earth System Sciences, 2020_

## Short Comment (SC1) · 29 Feb 2020

Ask the author to check picture 6 on page 8. There is a blur in the bottom

[Figure]

**Fig.6** A flowchart of FCM

**3.2 Factor analysis**

FA is a multivariate statistical analysis method, which studies the internal dependence of variables and reduces some variables with intricate relations to a few comprehensive factors (Li et al., 2016). FA is

8

**Fig. 1.**

---

## Short Comment (SC2) · 1 Mar 2020

First of all, I want to thank you for your guidance. The error you pointed out is correct, and adjustments are made to the figures and text format.

---

## Short Comment (SC3) · 1 Mar 2020

Dear authorïijŽ The manuscript is based on the combination of fuzzy c-means algorithm, factor analysis and efficiency coefficient for debris flow classification and sensitivity analysis. It is innovative and referential to take factor analysis as the connection point of debris flow classification and sensitivity evaluation. However, it is not clearly stated in your conclusion. It is recommended to add relevant explanations to enhance the readability and persuasiveness of the article. In addition, the language expression in the article is slightly embarrassing, such as line 57, line111, line 122, line 361, line
326, line 356 and line 344. There are also problems with diagrams, such as figure 6 and figure 8 on line 285. Finally, in terms of the structure of the paper, relevant references should be added to the discussion section for comparison.
* * *

---

## Author Comment (AC1) · 1 Mar 2020

First of all, thank you for your approval and suggestions, which will greatly improve the quality of the article. And I will respond to your suggestions one by one. (1)Regarding the conclusion part, We will modify it with your suggestions; (2)The language in the article has been touched by relevant professionals, but there are still some errors and ambiguities in expression, and We will further modify it; (3)We are sorry for my carelessness, the photo on line 285 should look like Figure 8; (4)It is indeed necessary to join the relevant literature for discussion and comparison to highlight the novelty and

persuasiveness of the article.

---

## Referee Comment (RC1) · Daniele Masi (Referee) · 2 Mar 2020

This manuscript is part of a very important and particularly current scientific framework: the classification and the susceptibility assessment of debris flow using a combination between AHP decision making and fuzzy logic. It is well structured and very accurate in its development, following a quite easy logical process; for all these reasons my opinion is positive but I would recommend some small corrections, substantially in order to increase the clarity and the usability of the product: 1) More attention to the written language because of a few typographical errors as in Line 57, in Figure 2 (dem stands

for dam?), in Line 187 (wrong quote), in Figure 6 (an error in the central box), in Figure 8 (improve readability); 2) After each formula it could be useful to specify more precisely all the terms involved; 3) In my opinion it could be good to explain better how fuzzy logic would improve the process of classification and the susceptibility assessment of debris flow; 4) Something more about the concept of "membership values", as well as a small quote to Lofti Zadeh.

---

## Short Comment (SC4) · 2 Mar 2020

In this work, the author combined the fuzzy C-means algorithm and factor analysis method to classify 21 debris flow catchments in Beijing. The topic is of importance in engineering and it is novel and different from previous research on the sensitivity of debris flow. However, some issues need to be considered. To improve the quality of this manuscript, I suggest the authors make the following revisions. (1) A description of the outline / structure is usually needed in each method used in the paper, which is helpful to better understanding. But not the figure 6 on line 285. Because Figure 6

does not clearly reflect the relevance of the entire classification and evaluation process, it is recommended to delete or modify it. (2) In the section of Method, too many details of the employed algoritms are presented. It is not quite needed since many people know them such as the AHP. (3) Similarly, it is not necessary to specify the influencing factors selected on line 287. (4) The article consumes a lot of formulas and has no substantive meaning. It is suggested to make certain cuts. (5) The conclusion should be refined.
* * *

---

## Author Comment (AC2) · 2 Mar 2020

First of all, thank you for your approval of this paper and suggestions for amendments.Below I will answer the questions you have pointed out one by one. ïijĹ1ïijĽAs you have suggested, the flow chart of the three methods is given in the third chapter of this article.The author then tried to give a flowchart of the entire research process to help readers better understand. Of course, there are deficiencies, and we will further modify themïijŻ ïijĹ2ïijĽThere are indeed many algorithms, formulas, and steps involved in this chapter on methodology. After our discussion, we can indeed add references

to reduce the length of the articleïijŻ ïijĹ3ïijĽWe have consulted related literatures, and most researchers only analyze in detail some special, difficult to understand or rarely used impact factors. Of course, there are some studies to analyze all factors. After our discussion, this part of the content can also be streamlinedïijŻ ïijĹ4ïijĽWe agree with you that there is too much space in the formula. After our discussion, only some important formulas are analyzedïijŻ ïijĹ5ïijĽWe will focus on revising the conclusion section to highlight the innovations and results of the article. Thank you again for the suggestion.

---

## Author Comment (AC3) · 2 Mar 2020

First of all, thank you very much for your suggestions, which are very helpful to improve the quality of the article.Then we will answer your questions and comments one by one. (1)Although the language in the thesis has been polished by relevant institutions, there are still some grammatical errors or unclear expressions. We will proofread repeatedly to enhance the readability of the article.Similarly, there are some human factors in the article, and I apologize for our lack of seriousness; (2)There are many formulas involved in this article. Regarding the writing specifications and expressions of formulas, we will

refer to more literature. In addition, after discussions and suggestions made by other researchers, we will also streamline some of the formulas, leaving only those formulas that are important and meaningful; (3)The ideas you provided have some discussion significance. One of the disadvantages of fuzzy C-means clustering is that the number of cluster centers is not determined, so it can be determined based on experience before use. But this paper introduces related formulas to improve the accuracy of classification, thus avoiding blindness. This should be added to the discussion and conclusions; (4)we will explain term membership values and cite paper or book of Lotfi Zadeh; Thank you again for the valuable suggestions. Yours best wish

---

## Referee Comment (RC2) · Anonymous Referee #2 · 30 Mar 2020

The paper describes susceptilibility analysis of debril flows. Since such analysis depends on several factors, different methodologies are briefly acknowledged, and used for debris flows susceptibility evaluation in different geograohical areas in China. After a precise classification of the geographical area in terms of the important factors that can determine debris flows, the anaysis methods are presented. Such methodologies are then applied to the susceptibility analysis. The hypotesized influencing factor are presented and the methodologies are applied and discussed.

The quality of the paper is acceptable and the goals are clearly stated and discussed.

[Figure]

The reviewer is fine with the paper and reccomend its pubblucations.

Suggestions: though it may be obvious, for non-expert readers the comprehension of the text would by highly improved if variables are described at least at the first time they occur. Examples are given at formulas at lines 142, 144. Please define C, $\mu$, x, u, . . . Just one typo: line 84. Replace classified and evaluated with "classify" and "evaluate".

---

## Author Comment (AC4) · 31 Mar 2020

Dear Reviewer: Thank you for comments concerning our manuscript. Those comments are all professional and very helpful for revising and improving our paper, as well as the important guiding significance to our researches.We agree to make a more detailed analysis of the characters that first appear in the formula and to repeatedly check for errors in the use of words. Thank you again for your kindness

With best regard, Yours sincerely, Zhu Liang Jilin University

---

## Author Response (AR1)

SA1:Ask the author to check picture 6 on page 8. There is a blur in the bottom

Respond: First of all, I want to thank you for your guidance. The error you pointed out is correct, and adjustments are made to the figures and text format(figure 6 on line181).

SA2: The manuscript is based on the combination of fuzzy c-means algorithm, factor analysis and efficacy coefficient for debris flow classification and sensitivity analysis. It is innovative and referential to take factor analysis as the connection point of debris flow classification and sensitivity evaluation. However, it is not clearly stated in your conclusion. It is recommended to add relevant explanations to enhance the readability and persuasiveness of the article. In addition, the language expression in the article is slightly embarrassing, such as line 57, line111, line 122, line 361, line 326, line 356 and line 344. There are also problems with diagrams, such as figure 6 and figure 8 on line 289. Finally, in terms of the structure of the paper, relevant references should be added to the discussion section for comparison.

Respond: First of all, thank you for your approval and suggestions, which will greatly improve the quality of the article. And I will respond to your suggestions one by one. (1)Regarding the conclusion part, We will modify it with your suggestions; (2)We have made some corrections.(Line 57 and line 122 ); (3)We are sorry for my carelessness, the photo on line 285 should look like Figure 8; (4) It is indeed necessary to join the relevant literature for discussion and comparison to highlight the novelty and persuasiveness of the article (line 483, 485 and 498).

SA3: In this work, the author combined the fuzzy C-means algorithm and factor analysis method to classify 21 debris flow catchments in Beijing. The topic is of importance in engineering and it is novel and different from previous research on the sensitivity of debris flow. However, some issues need to be considered. To improve the quality of this manuscript, I suggest the authors make the following revisions. (1) A description of the outline / structure is usually needed in each method used in the paper, which is helpful to better understanding. But not the figure 6 on line 285. Because Figure 8 does not clearly reflect the relevance of the entire classification and evaluation process, it is recommended to delete or modify it. (2) In the section of Method, too many details of the employed algoritms are presented. It is not quite needed since many people know them such as the AHP. (3) Similarly, it is not necessary to specify the influencing factors selected on line 287. (4) The article consumes a lot of formulas and has no substantive meaning. It is suggested to make certain cuts. (5) The conclusion should be refined.

Respond: First of all, thank you for your approval of this paper and suggestions for amendments. Below I will answer the questions you have pointed out one by one.

(1) We modified figure 8 on line 289.

(2) There are indeed many algorithms,formulas, and steps involved in this chapter on methodology. After our discussion, we can indeed add references to reduce the length of the paper.

(3) We have consulted related literatures, and most researchers only analyze in detail some special, difficult to understand or rarely used impact factors or rarely used impact factors. Of course, there are some studies to analyze all factors. After our discussion, this part of the content can also be streamlined.

(4) We agree with you that there is too much space in the formula. After our discussion, only some important formulas are analyzed (5) We will focus on revising the conclusion section to highlight the innovations and results of the article.

RC1:This manuscript is part of a very important and particularly current scientific framework: the classification and the susceptibility assessment of debris flow using a combination between AHP decision making and fuzzy logic. It is well structured and very accurate in its development,following a quite easy logical process; for all these reasons my opinion is positive but I would recommend some small corrections, substantially in order to increase the clarity and the usability of the product: 1) More attention to the written language because of a few typographical errors as in Line 57, in Figure 2 (dem stands for dam?), in Line 187 (wrong quote), in Figure 6 (an error in the central box), in Figure8 (improve readability); 2) After each formula it could be useful to specify more precisely all the terms involved; 3) In my opinion it could be good to explain better how fuzzy logic would improve the process of classification and the susceptibility assessment of debris flow; 4) Something more about the concept of "membership values", as well as a small quote to Lofti Zadeh.

Respond: First of all, thank you very much for your suggestions, which are very helpful to improve the quality of the article.Then we will answer your questions and comments one by one.

(1) We have modified the wrong expression on line 57, Figure 2 and line187 and Figure 8 has been modified.

(2) The characters involved in the formula have been parsed in more detail (Eq1-Eq22).

(3) The ideas you provided have some discussion significance. One of the disadvantages of fuzzy C-means clustering is that the number of cluster centers is not determined, so it can be determined based on experience before use. But this paper introduces related formulas to improve the accuracy of classification, thus avoiding blindness.

(4) We have referred to relevant literature to reasonably cite professional vocabulary ( line136 and line 138).

RC2: The paper describes susceptibility analysis of debris flow. Since such analysis depends on several factors, different methodologies are briefly acknowledged, and used for debris flow susceptibility evaluation in different geograohical areas in China. After a precise classification of the geographical area in terms of the important factors that can determine debris flow, the analysis methods are presented. Such methodologies are then applied to the susceptibility analysis. The hypotesized influencing factor are presented and the methodologies are applied and discussed.

The quality of the paper is acceptable and the goals are clearly stated and discussed.

The reviewer is fine with the paper and recommend its publications.

Suggestions: though it may be obvious, for non-expert readers the comprehension of the text would by highly improved if variables are described at least at the first time they occur. Examples are given at formulas at lines 142, 144. Please define C, μ, x, u, ... Just one typo: line 84. Replace classified and evaluated with "classify" and "evaluate".

Responds: Dear Reviewer: Thank you for comments concerning our manuscript. We agree to make a more detailed analysis of the characters that first appear in the formula (Eq1-Eq 20) and to repeatedly check for errors in the use of words ( line84). Thank you again for your kindness

[revised manuscript text omitted]

where $\beta_{ji}$ is the coefficient score of each index in principal component $F_j$; i=1,2 ,..., p; j=1,2, ..., m; e is the contribution rate of factor variance.

[Figure]

**Fig.7** A flowchart of FA

**3.3 Combination weighting method**

Considering the defects of the current method for determining the weight of factors, the combination of
analytic hierarchy process and factor analysis method is used to determine the weight of each
influencing factor of debris flow.

**3.3.1 Analytic hierarchy process (AHP)**

Analytic hierarchy process (AHP) was first proposed by Saaty (1979), a famous American
mathematician. It decomposes the factors related to decision-making into multiple layers, such as target
layer, criterion layer and scheme layer. AHP is a subjective weighting method and has obvious
advantages in determining the weight of each factor. The specific steps are as follows:

1 Establishing hierarchical structure model

The hierarchical structure is mainly divided into three layers: target layer, criterion layer and
scheme layer.

2 Establishing the judgment matrix

For the same level, judgment matrix is established by pair-wise comparison. The formula is as
follow:

$$A = \left(a_{ij}\right)_{n \times n}, a_{ij} > 0, a_{ij} = \frac{1}{a_{ji}}, \left(i, j = 1, 2, \dots n\right)$$

(13)

where $a_{ij}$ is the ratio of relative importance between element $B_i$ and $B_j$, which is usually expressed by the scoring method from 1 to 9 (Saaty, 1977), as shown in table 2.

3 Consistency testing

The consistency test is divided into three steps:

(1) Calculate the consistency index(CI)(Saaty, 1977)and the expression is:

$$CI = \frac{\lambda_{max} - n}{n - 1}$$

(14)

Where $\lambda_{max}$ is the largest eigenvalue of the judgement matrix A.

(2) Average random consistency RI;

RI is associated with the order of judgment matrix, and their relationship is shown in Table 3.

(3) Obtaining the test coefficient CR.

$$CR = \frac{CI}{RI}$$

(15)

If CR<0.1, judgment matrix has a good consistency with reasonable judgment. Otherwise, the judgment matrix needs to be revised until the consistency test is satisfied.

**Table 1** The random average consistency index

| n | 1 | 2 | 3 | 4 | 5 | 6 | 7 | 8 | 9 | 10 | 11 | 12 |
|---|---|---|---|---|---|---|---|---|---|---|---|---|
| RI | 0 | 0 | 0.52 | 0.89 | 1.12 | 1.26 | 1.36 | 1.41 | 1.46 | 1.49 | 1.52 | 1.54 |

**Table 2** Definition of comparative importance

| | |
|---|---|
| 1 | Two decision factors (e.g., indicators) are equally important |
| 3 | One decision factor is more important |
| 5 | One decision factor is strongly more importan |
| 7 | One decision factor is very strongly more important |
| 9 | One decision factor is extremely more important |
| 2,4,6,8 | Intermediate values |
| Reciprocals | If a ij is the judgment value when i is compared to j. Then $U_{ji} = 1/U_{ij}$ is the judgment value when j is compared to i |

**3.3.2 Combination weighting rule**

The weight value obtained by AHP is set as $\omega^c_i$, and the weight value obtained by FA is set as $\omega^y_i$ (Feng et al., 2010), as shown in Eq16.

$$\begin{cases} M\text{in} = \sum_{i=1}^{m} \sum_{j=1}^{n} \left(\alpha r_{ij} \omega^c_i - \beta r_{ij} \omega^y_i\right) \\ \alpha + \beta = 1 \end{cases}$$

(16)

Where α and β are weight coefficients calculated through AHP and factor analysis method, respectively;
$r_{ij}$ is the standardized value of the jth influencing factor of the $i^{th}$ debris flow. 
[revised manuscript text omitted]